# OpenReview forum: "Security Tensors as a Cross-Modal Bridge: Activating Text-Aligned Safety to Vision in LVLMs"
_ICLR.cc/2026/Conference — Submitted to ICLR 2026_

### Official Review · Reviewer_ZYrM · 2025-10-29

**Soundness:** 2
**Presentation:** 2
**Contribution:** 2
**Rating:** 4
**Confidence:** 3

**Summary:**

This paper introduces Security Tensors, a novel, parameter-free method to bridge the cross-modal safety gap in LVLMs. By optimizing input perturbations on a curated dataset, the method activates the language module's inherent text-aligned safety mechanisms to handle harmful visual inputs, without requiring any changes to the model's weights.

**Strengths:**

1. The method shows significant safety improvements and strong generalization across multiple LVLMs, with minimal degradation of performance on benign tasks.

2. The experimental setup is described in a detailed and thorough manner.

**Weaknesses:**

1. Lack of Analysis on Model Scaling Effects. A systematic study, for instance across the Qwen3-VL family (e.g., 2B, 4B, 8B, 30B), would be necessary to validate the scalability and robustness of this approach, thereby defining the boundaries of its applicability.

2. Insufficient Consideration of Adaptive Attacks. Without this analysis, the true security guarantees of the method in a dynamic, real-world adversarial environment remain unverified.

3. Inherent Dependency on the Base Model's Safety Alignment. This limits its applicability and positions it as an incremental improvement for already well-aligned models, rather than a universal solution for securing any given LVLM. Investigating whether combining this approach with pre-processing defenses could achieve a more robust "1+1 > 2" outcome would have been a valuable analysis, positioning the work within a more practical, defense-in-depth framework.


4. Absence of Computational Analysis. Without these measurements, the claim of being a practical and lightweight solution is unsubstantiated, especially for applications where low latency is critical.

**Questions:**

Please refer to the questions raised in the Weaknesses section above.

---

> ### Author Response · Authors · 2025-11-20
> **Response to Reviewer  ZYrM - Part 1**
>
> We sincerely appreciate the insightful feedback provided and offer the following detailed responses to address each of the concerns raised:
>
> **[Weakness 1]. Lack of Analysis on Model Scaling Effects. A systematic study, for instance across the Qwen3-VL family (e.g., 2B, 4B, 8B, 30B), would be necessary to validate the scalability and robustness of this approach, thereby defining the boundaries of its applicability.**
>
> ---
>
> Thank you for the valuable comment on model scaling effects. During the rebuttal phase, we conducted additional experiments on a family of Google's Gemma-3 pretrained multimodal models (gemma-3-4B-pt / 12B-pt / 27B-pt). Our results demonstrate that the proposed method exhibits consistent performance across different model scales. Below we summarize (1) the scaling results, and (2) why we chose Gemma-3-pt rather than the Qwen3-VL family for this analysis.
>
> **(1) Scaling experiments on Gemma-3-pt models of different sizes**
>
> To directly address the question of scalability and robustness, we trained visual-side and text-side security tensors ($\delta_v$ and $\delta_t$) on gemma-3-4B-pt, gemma-3-12B-pt, and gemma-3-27B-pt under exactly the same training setup as for LLaMA-3.2-11B-Vision in the main paper. We report:
>
> - The average Harmless Rate (HR) on malicious inputs (seen and unseen categories)
> - The False Rejection Rate (FRR) and MM-Vet score on benign inputs, to assess harmlessness and utility
>
> A summary of the main trends is shown below (full numeric results are provided in the revised appendix):
>
> | | | HR (Seen) | HR (Unseen) | FRR | MM-Vet |
> | - | - | - | - | - | - |
> | gemma-3-4B-pt | base | 17.6 | 18.2 | 0.25 | 49.3 |
> | | ST-$\delta_t$ | 72.3 | 74.0 | 0.75 | 49.2 |
> | | ST-$\delta_v$ | 74.4 | 76.8 | 1.50 | 48.9 |
> | gemma-3-12B-pt | base | 25.8 | 25.0 | 0.50 | 57.6 |
> | | ST-$\delta_t$ | 80.8 | 83.6 | 1.00 | 57.0 |
> | | ST-$\delta_v$ | 82.2 | 83.5 | 1.25 | 57.2 |
> | gemma-3-27B-pt | base | 28.4 | 27.8 | 0.25 | 65.2 |
> | | ST-$\delta_t$ | 85.7 | 87.2 | 0.50 | 65.3 |
> | | ST-$\delta_v$ | 86.8 | 88.4 | 0.75 | 64.7 |
>
>
> Table 1. Scaling results on gemma-3-pt models of different sizes: average HR on seen categories, HR on unseen categories, FRR on benign inputs, and MM-Vet scores, for the base model and the models equipped with text-side (ST-$\delta_t$) or visual-side (ST-$\delta_v$) security tensors.
>
> Overall, these results indicate that, across gemma-3-pt models from 4B to 27B, the security tensors consistently and substantially improve refusal behavior on harmful visual inputs (both seen and unseen categories), while maintaining benign performance (FRR and MM-Vet) at acceptable levels. This provides systematic evidence that our method scales reliably across LVLMs of different sizes.
>
> **(2) Why we use the Gemma-3-pt family rather than Qwen3-VL for scaling analysis**
>
> We thank the reviewer for suggesting a scaling study on the Qwen3-VL family (2B / 4B / 8B / 30B). Our method, however, is specifically designed for the following setting: the language modality has already undergone substantial safety alignment, while the visual modality has *not* received an independent visual safety alignment. In this “alignment mismatch” regime, harmful content appearing only in the image fails to reliably trigger the existing text-side safety mechanisms. Our security tensors are precisely meant to address this case by using lightweight input-level controls to activate and reuse the aligned text-side safety for the visual modality, without modifying LVLM parameters.
>
> However, the Qwen3-VL-Instruct models are multimodally pretrained and then further aligned with dedicated, high-quality *visual* safety procedures. Our tests in rebuttal indicate that their visual safety is already very strong out of the box. Applying security tensors on top of these models would mainly study “further boosting an already heavily visually aligned LVLM,” and could blur our core message by suggesting that our method is primarily an add-on for highly secured commercial systems, rather than a mechanism for activating latent text-side safety in under-protected visual modalities.
>
> By contrast, the Gemma-3-pt pretrained multimodal models are more consistent with the models used in our main experiments (LLaVA, Qwen-VL-Chat, LLaMA-3.2-Vision): they have a strong language backbone with safety alignment, are connected to a visual encoder, but lack a dedicated, separate visual safety alignment stage.
>
> In summary, the Gemma-3-pt scaling experiments show that our security tensors maintain strong safety gains and stable benign behavior across small to larger LVLMs in the regime our method targets. We have added this part of the experiment to the Appendix A.1.9 of the revised version. Thanks again for your comments!

---

> ### Author Response · Authors · 2025-11-20
> **Response to Reviewer  ZYrM - Part 2**
>
> **[Weakness 2].  Insufficient Consideration of Adaptive Attacks. Without this analysis, the true security guarantees of the method in a dynamic, real-world adversarial environment remain unverified.**
>
> ---
>
> Thank you for your comment on adaptive attacks.  We show the analysis as below:
>
> **(1) Threat model in this work: static safety under non-adaptive malicious distributions**
>
> First, we clarify the threat model and evaluation scope in this paper: Similar to many existing LVLM safety works, we focus on non-adaptive malicious input distributions and evaluate static robustness:
>
> * Harmful samples are drawn from independently constructed datasets where malicious information is embedded in images, covering both seen and unseen harmful categories.
> * The attacker does not know the exact value of the security tensor $\delta$ and interacts with the LVLM through a standard inference interface.
>
> Under this threat model, our experiments on multiple models and multiple harmful categories (including unseen ones), together with the layer-wise safety activation analyses (section 5 of the main paper), show that a fixed $\delta$ can effectively activate and reuse the language module’s existing safety capability in the most mainstream setting.
>
> **(2) Initial exploration toward “stronger / partially adaptive” attacks**
>
> During the rebuttal phase, we conducted a preliminary analysis to probe the behavior of $\delta$ under stronger, partially adaptive attacks.
>
> On the visual side, we attempted to train an “inducing” universal vector $\delta_{jb}$ whose objective is to push the model toward affirmative, jailbreak-style responses, i.e., nudging outputs toward openings like “Sure, the answer is …”, thereby behaviorally counteracting the refusal tendency induced by the safety vector. At test time, we then applied $\delta_{safe}$ (our learned safety vector) and $\delta_{jb}$ together and evaluated the resulting behavior.
>
> Empirically, we observed: With $\delta_{jb}$ applied alone, the model exhibits degraded safety performance on harmful inputs. When $\delta_{safe}$ and $\delta_{jb}$ are linearly combined at inference, the model’s outputs often become unstable: instead of cleanly reverting to fluent harmful answers, the generations frequently exhibit semantic incoherence, contradictions, or broken sentences.
>
> This indicates that a naive “second universal perturbation” that attempts to cancel $\delta_{safe}$ does not succeed as an attack: instead of restoring unsafe behavior in a controlled way, it severely degrades the model’s behavior on both harmful and benign inputs. In practice, the model’s responses become unstable and low-quality overall, so the attacker fails to reliably obtain coherent unsafe answers, and the attempted bypass should be regarded as unsuccessful.
>
> **(3) Future directions**
>
> From a broader robustness perspective, prior work has shown that: any fixed-form defense mechanism—whether a fixed system prompt, a fixed filtering rule, a fixed safety classifier, or a fixed universal defense perturbation like our $\delta$—can in principle be broken by a sufficiently strong and fully adaptive adversary. This is a well-known limitation in the adversarial robustness literature, and our method is not exempt from it.
>
> Importantly, within this context, our approach has some advantages and extension opportunities that make it a promising building block in an adaptive setting:
>
> * Extremely low training/update cost: Because $\delta$ has very few parameters and is trained without touching the LVLM weights, it can be retrained or updated quickly when new attack patterns are discovered, effectively serving as a low-cost “safety patch” compared to re-aligning the entire LVLM.
> * Potential for ensembles and randomized defenses: It is natural to extend our approach to a set of safety vectors (${\delta^{(1)},\dots,\delta^{(M)}}$) and, for example, randomly select or conditionally switch between them at inference time. Such randomization could make it harder for an adaptive attacker to overfit to a single, fixed defense direction.
>
> In summary, our experiments demonstrate that a fixed security tensor yields consistent and significant safety gains across models and harmful categories. For stronger adaptive attacks, our preliminary analysis shows that simple adversarial overlays cannot reliably bypass our defense without severely degrading utility. Given the extremely small parameter size and easy updatability of the security tensors, we believe extending this mechanism to rapid patching and multi-tensor randomization for more complex adaptive settings is a highly promising direction for future work.  Thanks again for your comments.

---

> ### Author Response · Authors · 2025-11-20
> **Response to Reviewer  ZYrM - Part 3**
>
> **[Weakness 3]. Inherent Dependency on the Base Model's Safety Alignment. This limits its applicability and positions it as an incremental improvement for already well-aligned models, rather than a universal solution for securing any given LVLM. Investigating whether combining this approach with pre-processing defenses could achieve a more robust "1+1 > 2" outcome would have been a valuable analysis, positioning the work within a more practical, defense-in-depth framework.**
>
> ---
>
> Thank you for the thoughtful comment.
>
> **(1) Positioning of our method: reusing text-side safety for vision, rather than a tiny tweak on “already well-aligned” models**
>
> We would first like to clarify that there is a misunderstanding regarding your perspective. The goal of this work is not to make a small incremental improvement on LVLMs that are already well-aligned in the visual modality. Instead, we deliberately focus on a very common but under-explored real-world situation:
>
> The language backbone has already undergone costly and careful safety alignment on the text side, but this safety capability is not activated or reused for visual inputs. In contrast, the visual branch often has almost no reliable safety protection: when harmful information appears only in the image, the existing safety mechanism is frequently bypassed and the model continues along the normal answering path.
>
> Our security vectors are designed exactly around this “mismatch” scenario. We use lightweight, input-level control to activate and transfer the already aligned text-side safety capability into the visual modality.  In this sense, the method is not a minor refinement on an already safe visual system; rather, it aims to bring the visual modality from “almost no safety” up to a level where it can effectively reuse the strong text-side alignment that already exists, avoiding the need to retrain an entirely new and expensive safety stack for vision.
>
> As also noted by reviewer 5hyW, this is the first work that explicitly formulates “reusing text-side safety mechanisms for the visual modality” as a central problem. Beyond proposing the mechanism, we also conduct “safety layer activation” analyses and related experiments to show how the security vectors re-activate layers that were originally only sensitive to harmful text. This gives the approach a more interpretable and mechanism-grounded characterization, rather than treating it as a black-box improvement.
>
> **(2) Combination with pre-processing defenses: a “1+1 > 2” defense-in-depth effect**
>
> During the rebuttal phase, we conducted additional experiments to investigate how our method can be combined with pre-processing defenses.  Concretely, we chose ECSO [1] as a representative pre-processing defense. ECSO uses an image-to-text transformation to convert visual content into descriptive text, which is then fed into the LVLM as an additional input. This is a pre-processing defense, whereas our security vectors act as an internal control during the LVLM’s forward pass. By design, these two mechanisms are complementary and can be composed in sequence.
>
> Under the same experimental setup as in the main paper, we evaluated the Harmless Rate (HR) on the malicious test set for different combinations. The results are:
>
>
>
> |                      | Base model | ECSO | $\delta_v$ | $\delta_t$ | ECSO + $\delta_v$ | ECSO + $\delta_t$ |
> | -------------------- | ---------- | ---- | ---------- | ---------- | ----------------- | ----------------- |
> | LLaVA-1.5            | 7.7        | 37.3 | 49.5       | 52.0       | 59.2              | 55.6              |
> | Qwen-VL-Chat         | 19.0       | 51.2 | 64.5       | 65.6       | 72.3              | 74.0              |
> | LLaMA-3.2-11B-Vision | 24.8       | 68.9 | 84.2       | 81.9       | 88.7              | 89.4              |
>
> Table 2. Harmless Rate (HR, %) of different LVLMs under pre-processing defense (ECSO), security vectors ($\delta_v$ / $\delta_t$), and their combinations. Higher is better.
>
> We observe that, for all three models, combining ECSO with our security vectors (ECSO + $\delta_v$ or ECSO + $\delta_t$) consistently yields higher HR than using ECSO alone or using the security vectors alone.
>
> Taken together, these results support two main points: (i) our method is best viewed as a mechanism for reusing and activating existing text-side safety alignment in the visual modality, rather than as a minor add-on for already well-secured LVLMs, and (ii) it integrates naturally with pre-processing defenses and can provide additional gains in a defense-in-depth framework. We have added this part of the analysis to Appendix A.1.8 of the revised version.
>
> [1] Gou et al., “Eyes closed, safety on: Protecting multimodal LLMs via image-to-text transformation.” ECCV 2024.

---

> ### Author Response · Authors · 2025-11-20
> **Response to Reviewer  ZYrM - Part 4**
>
> **[Weakness 4]. Absence of Computational Analysis. Without these measurements, the claim of being a practical and lightweight solution is unsubstantiated, especially for applications where low latency is critical.**
>
> ---
>
> Thank you for raising this question! Our claim of being “lightweight and practical” is based on two aspects: (1) only a very small number of additional parameters are trained, and (2) the inference-time latency overhead is negligible. We detail both below.
>
> **(1) Training-time parameter and computation overhead**
>
> Our method does not finetune the LVLM itself. Instead, we introduce two small sets of trainable tensors in the input space independently: a visual-side security tensor $\delta_v$ and a text-side security tensor $\delta_t$. For example, in the case of LLaVA-1.5-7B:
>
> - Visual-side security tensor $\delta_v$: added directly to the pre-processed image tensor, with shape $3 \times 336 \times 336$, corresponding to about 338688 parameters.
> - Text-side security tensor $\delta_t$: implemented as 100 virtual tokens, each of dimension 4096, for a total of $100 \times 4096 = 409600$ parameters.
>
> Compared to a 7B-parameter LVLM, these two components are on the order of $10^{-3}$of the total parameters (approximately 0.0048% and 0.0059%, respectively). For other LVLMs (LLaMA-3.2-Vision, Qwen-VL-Chat, etc.), the relative sizes of $\delta_v$ and $\delta_t$ are similarly tiny, as summarized in Table 3:
>
> |                   | LLaVA-1.5 | Qwen-VL-Chat | LLaMA-3.2-11B-Vision |
> | ----------------- | --------- | ------------ | -------------------- |
> | $\delta_v$ / LVLM | 0.0048%   | 0.0063%      | 0.016%               |
> | $\delta_t$ / LVLM | 0.0059%   | 0.0043%      | 0.012%               |
>
> Table 3. Relative parameter size of visual-side ($\delta_v$) and text-side ($\delta_t$) security tensors compared to the base LVLM parameters.
>
> During training, we update only these two tensors; no gradients are applied to any of the LVLM’s own weights. As a result, the additional memory and compute overhead is comparable to training a very small LoRA module, and is far lower than full-model finetuning or large-PEFT configurations.
>
> **(2) Inference-time latency overhead**
>
> The visual tensor $\delta_v$ is added element-wise to the pre-processed image tensor. This operation does not change the tensor shape (resolution, channels, or number of visual tokens all remain the same), nor does it introduce new operators or tokens. The FLOPs of the visual encoder are determined by the tensor shape, not by the specific pixel values. Therefore, adding $\delta_v$ does not change the computational complexity of the visual encoding stage. On a single A100 GPU, we repeatedly ran inference on the same image with and without $\delta_v$ and observed that the difference in average latency per run lies within measurement noise, indicating that the practical overhead of $\delta_v$ at inference time is negligible.
>
> The text-side tensor $\delta_t$ is equivalent to prepending some additional virtual tokens on the text side. In typical single-image multimodal usage, existing LVLMs already process hundreds of visual tokens (e.g., around 500–600 visual tokens for a single image in our tested LVLMs), plus the user’s textual query. Many deployed models also use very long system prompts in practice; for example, public examples of GPT-4/4o-style system prompts often contain several hundred to over 500–800 tokens.
>
> Against this background, adding some extra tokens corresponds to only a modest constant-factor increase in context length. In our A100-based tests on the same batch of benign and malicious inputs, we measured only a slight increase in average inference latency when $\delta_t$ is enabled, and the total latency remains in the same order of magnitude as the base LVLM. This overhead is much smaller than that of approaches that introduce additional safety classifiers or require multiple forward passes.
>
> In summary, during training we optimize only a very small number of input-level parameters, which significantly reduces computation and storage requirements compared to full-model finetuning. During inference, the visual-side security tensor does not alter the computation graph at all, and the text-side tensor only adds a small constant-factor increase in context length, with empirically very limited latency overhead. In the revised version, we have added content in  A.1.8 reporting the above computational analysis to make this “lightweight training” aspect explicit. Thanks again for your comments.
>
>
>
> ------
>
> **In conclusion, we want to express our gratitude for the exceptionally positive impact your comments had on our contribution. If there are any remaining questions or issues, please let us know—we are happy to assist at any time. We also hope that our responses effectively address any questions you may have had and perhaps sway your consideration more favorably towards accepting the paper. Sincerely thank you again!**

---

### Official Review · Reviewer_2kK7 · 2025-10-29

**Soundness:** 2
**Presentation:** 2
**Contribution:** 3
**Rating:** 6
**Confidence:** 4

**Summary:**

This paper proposes security tensors, which are trainable input-level perturbations added during inference through either the text or vision pathway of an LVLM, to activate the model’s existing textual safety mechanisms when it encounters harmful visual inputs.

**Strengths:**

- Works as a plug-in without modifying LVLM weights
- Large HR improvements across three LVLMs; meaningful generalization to unseen harmful categories.
- Low FRR, small MM-Vet impact
- Layer-wise alignment with prior “safety layers” strengthens the causal story.

**Weaknesses:**

- The N – (N + δ) pairs in Figure 3 show considerable differences, especially in the deeper layers. The gap between N – (N + δ) and M – (M + δ) is not substantially different, so how can it be demonstrated that δ has only a small impact on benign image–text inputs?

- Please report the actual number of samples in the test set described in L298, including both the Seen-category and Unseen-category subsets. To my understanding, the scales of VLGuard and MM-SafeBench are not very large, which raises concerns about the adequacy of the evaluation setting. If I have misunderstood, please clarify.

- Training uses only 4 harmful categories and 1000 samples. I would like to know how the number 1000 was determined—was there an ablation study on this number? Would δ learned from a larger dataset yield better performance?

- In L209, the paper mentions “refusal templates,” but their scale and diversity are not discussed. This raises concern that the model’s outputs may become formulaic refusals. Have the authors considered distinguishing between semantic safety and template matching? An analysis of output diversity versus the variation among refusal templates could provide stronger evidence that “the model learns the semantic intent of refusal rather than memorizing superficial token patterns.”

- Since δ is fixed, an adaptive adversary could design counter-perturbations. Adding evaluations under adaptive adversarial attacks would better demonstrate δ’s robustness in unseen scenarios.

- How is the Harmless Rate (HR) in L298 evaluated—by matching refusal keywords? This needs clarification. If HR is based on refusal keyword matching, I would be curious to see results under semantic refusal detection.

- Do the results hold for fine-grained harmful categories (e.g., medical self-harm paraphernalia, non-gory violence, subtle hate symbols) or for text-in-image cases?

**Questions:**

See Weaknesses

---

> ### Author Response · Authors · 2025-11-20
> **Response to Reviewer  2kK7 - Part 1**
>
> We sincerely thank you for your insightful and constructive comments! Your valuable feedback is highly appreciated. In response, we are providing a comprehensive point-by-point clarification to address each of the comments you raised:
>
> **[Weakness 1] The N – (N + $\delta$) pairs in Figure 3 show considerable differences, especially in the deeper layers. The gap between N – (N + $\delta$) and M – (M + $\delta$) is not substantially different, so how can it be demonstrated that $\delta$ has only a small impact on benign image–text inputs?**
>
> ---
>
> Thank you for the careful reading of Figure 3 and for raising this point!
>
> First, at the behavioral level, we evaluate whether the security tensors harm the models’ benign capabilities. In the main paper, we already show on MM-Vet that adding the security tensors does not degrade general LVLM performance(table 1 in the manuscript). During the rebuttal phase, we further evaluate the impact of the security tensors on domain-specific reasoning abilities using MathVista and MMMU dataset. The additional results are shown in Table 1:
>
> | | Qwen-VL-Chat | | | LLaVA-1.5 | | | LLaMA-3.2-11B | | |
> | - | - | - | - | - | - | - | - | - | - |
> | | Base | ST-$\delta_v$ | ST-$\delta_t$ | Base | ST-$\delta_v$ | ST-$\delta_t$ | Base | ST-$\delta_v$ | ST-$\delta_t$ |
> | MMMU | 35.9 | 36.0 | 36.2 | 36.4 | 35.4 | 35.8 | 50.7 | 50.2 | 50.6 |
> | MathVista (testmini) | 36.2 | 35.5 | 35.9 | 27.6 | 27.5 | 28.0 | 51.5 | 50.5 | 51.2 |
>
>
> *Table 1. Performance of different LVLMs on mathematical reasoning (MathVista testmini) and multi-discipline commonsense reasoning (MMMU) benchmarks, with and without the proposed security tensors $\delta_v$ and $\delta_t$.*
>
> It shows that, in terms of actual model outputs, the impact of $\delta$ on benign image–text inputs is small and controlled, rather than causing any substantial degradation of the original abilities.
>
> Second, regarding Figure 3 itself, the N – (N + $\delta$) curve alone should not be interpreted as evidence of “large disruption” on benign inputs. This curve measures the layer-wise cosine similarity between the last-token hidden states of the same benign query, with and without the added prefix δ. In practice, any extra prefix—even a harmless and task-agnostic one—can naturally induce noticeable changes in high-dimensional hidden states, particularly in deeper layers.
>
> To make this explicit, we conducted a controlled experiment in the rebuttal phase by replacing $\delta_t$ with a neutral system prompt $P_{system}$ (e.g., “Please think carefully and answer this question…”), whose length is comparable to $\delta_t$ but whose content is a generic, non-safety-related instruction. We found that the curve for N – (N + $P_{\text{system}}$) also exhibits considerable differences in deeper layers. This indicates that a large portion of the observed representation drift is a generic effect of adding any additional prefix in an autoregressive decoder, rather than a $\delta$-specific “strong disturbance.” The corresponding N – (N + $P_{\text{system}}$) curves are provided at the anonymous link:
>  https://anonymous.4open.science/r/ICLR2026_Rebuttal_Security_Tensor-BB87/cos_sim_plot.png. Therefore, when we compare N – (N + $\delta$) and M – (M + $\delta$) together in Figure 3, both curves share the common effect of “adding a prefix and changing the autoregressive context,” and their gap largely cancels out this shared component.
>
> ---
>
> **[Weakness 2] Please report the actual number of samples in the test set described in L298, including both the Seen-category and Unseen-category subsets. To my understanding, the scales of VLGuard and MM-SafeBench are not very large, which raises concerns about the adequacy of the evaluation setting. If I have misunderstood, please clarify.**
>
> ---
>
> Thank you for the question. We are happy to clarify the exact number of test samples and how the Seen / Unseen categories are constructed.
>
> In our experiments, we intentionally combine VLGuard and MM-SafeBench to obtain a more comprehensive and sufficiently large malicious test set. Concretely:
>
> - From VLGuard, we merge the original training and test splits and select all samples whose “safe” attribute is False, resulting in 1,465 malicious examples.
> - From MM-SafeBench, we use all 5,040 malicious examples.
>
> The distribution over harmful categories is shown in Table 2:
>
> | | Bloody | Porn | Gesture | Gun | Political | Privacy | Racial | Others |
> | - | - | - | - | - | - | - | - | - |
> | num | 704 | 637 | 802 | 579 | 625 | 582 | 834 | 1742 |
>
> *Table 2. Number of malicious samples per harmful category in the combined evaluation set.*
>
> We believe that the resulting evaluation setting is adequate for reliably assessing the safety behavior of the models in our study, with substantial coverage in both Seen and Unseen categories. Thank you again for raising this point!

---

> ### Author Response · Authors · 2025-11-20
> **Response to Reviewer  2kK7 - Part 2**
>
> **[Weakness 3] Training uses only 4 harmful categories and 1000 samples. I would like to know how the number 1000 was determined—was there an ablation study on this number? Would $\delta$ learned from a larger dataset yield better performance?**
>
> ---
>
> Thank you for raising the question. Regarding the specific scale of “1000” samples, in our preliminary experiments we compared with multiple smaller configurations: for example, when we trained the security vectors on only partial SA/GB/TCB subsets (with total size < 1000), we observed that both the safety improvement (HR) and benign performance (FRR / MM-Vet) were clearly inferior to the current configuration, and the results exhibited larger variance. When the total number of training samples increased to around 1000, the metrics became much more stable and consistently improved across all three LVLMs. We also tried enlarging the SA and TCB sets beyond this scale, but did not observe clear additional benefits.  For this reason, we chose to report this “small but stable” configuration in the paper.
>
> As for whether using a larger training set would further improve the effect of $\delta$, our view is: the security vector is essentially a low-capacity universal control direction, which slightly shifts the decision boundary within the already existing safety subspace, rather than relearning the entire task distribution from scratch. Under such a low-capacity setting, once the current training data already covers the major harmful patterns and sufficiently representative benign counterparts, further enlarging the dataset is expected to bring mainly marginal gains in statistical stability, and is unlikely to change the overall conclusions qualitatively. We will add the discussion about the training sample size into the final version, thanks again for your comments.
>
>
>
> **[Weakness 4] In L209, the paper mentions “refusal templates,” but their scale and diversity are not discussed. This raises concern that the model’s outputs may become formulaic refusals. Have the authors considered distinguishing between semantic safety and template matching? An analysis of output diversity versus the variation among refusal templates could provide stronger evidence that “the model learns the semantic intent of refusal rather than memorizing superficial token patterns.”**
>
> ------
>
> Thank you for raising this concern about refusal templates and the risk of formulaic outputs. Our design in L209 is in fact intended to avoid exactly this “template memorization” behavior, and we are happy to clarify both the scale/diversity of the templates and how we distinguish semantic safety from superficial pattern matching.
>
> In the SA set, we do not use a single fixed sentence as the refusal label. Instead, we construct a pool of K = 10 diverse refusal prefixes, denoted as $\mathcal{Y}_{\text{reject}} $. These are short, stylistically different refusal-style openings such as “I cannot assist with this request.”, “I am not able to help with this.”, and similar variants. For each SA sample, we randomly sample one template from this pool as the target refusal response. The goal is to reactivate and reuse the LLM’s existing safety alignment capability, that is, to learn the semantic decision of “when to refuse”, rather than to force the model to memorize a specific string. This design implicitly requires the learned security tensors to generalize across multiple phrasings instead of overfitting to a single token pattern.
>
> We also ran an early variant that used only a single refusal prefix (K = 1). In that setting, training on SA converged faster, but the final behavior matched the reviewer’s concern:
> – The improvement in harmlessness (HR) on unseen categories was noticeably weaker.
> – The false rejection rate (FRR) on benign inputs increased significantly.
>
> Compared to our current K = 10 configuration, this K = 1 setup showed worse generalization and stronger signs of template-level overfitting, which is fully consistent with the concern that overly narrow templates encourage superficial pattern memorization. This empirical observation motivated our choice to use a more diverse template pool.

---

> ### Author Response · Authors · 2025-11-20
> **Response to Reviewer  2kK7 - Part 3**
>
> **[Weakness 5] Since $\delta$ is fixed, an adaptive adversary could design counter-perturbations. Adding evaluations under adaptive adversarial attacks would better demonstrate $\delta$’s robustness in unseen scenarios.**
>
> ---
>
> Thank you for the comment. We make the following explanation:
>
> **(1) Preliminary analysis with stronger / partially adaptive attacks**
>
> During the rebuttal phase, we conducted an exploration of stronger attacks to probe how $\delta$ behaves under more aggressive, partially adaptive conditions.
>
> On the visual side, we trained an “inducing” universal vector $\delta_{jb}$ whose objective is to push the model toward affirmative responses such as “Sure, the answer is …”, thereby behaviorally counteracting the refusal tendency induced by the safety vector. We then evaluated the model under the combined perturbation $\delta_{safe}$ + $\delta_{jb}$ at inference time. Empirically, we observed:
>
> * With $\delta_{jb}$ applied alone, the model exhibits degraded safety performance on harmful inputs.
> * When we apply $\delta_{jb}$ on top of $\delta_{safe}$, the outputs do not revert to a clean, fluent “helpful but harmful” state. Instead, the model’s responses often become unstable, semantically incoherent, or stylistically degraded.
>
> This indicates that a naive “second universal perturbation” that attempts to cancel $\delta_{safe}$ does not succeed as an attack: instead of restoring unsafe behavior in a controlled way, it severely degrades the model’s behavior on both harmful and benign inputs. In practice, the model’s responses become unstable and low-quality overall, so the attacker fails to reliably obtain coherent unsafe answers, and the attempted bypass should be regarded as unsuccessful.
>
> **(2) Future directions**
>
> From a broader robustness perspective, we acknowledge that any fixed defense mechanism—whether a fixed system prompt, fixed filtering rules, a fixed safety classifier, or a fixed universal defense perturbation like $\delta$—can in principle be broken by a sufficiently strong, fully adaptive attacker. This limitation is a well-known consensus in adversarial robustness research, and our method is not an exception in that regard.
>
> Within this context, our approach has a few properties that make it amenable to extension toward more adaptive settings:
>
> * Extremely low training and update cost.
>   Since $\delta$ has very low parameter count and is trained entirely in the input space, it can be retrained or patched quickly when new attack modes emerge, without touching the base LVLM weights. Compared to re-aligning the entire LVLM, this greatly lowers the cost of iterative hardening.
>
> * Natural extensions to ensembles and randomization.
>   It is straightforward to extend our framework to learn a set of security tensors {$\delta$^(1), …, $\delta$^(M)} and either randomly select one at inference time or switch by scenario, thereby increasing the difficulty for an attacker to overfit to a single fixed defense direction. This points to a promising direction for building more robust, defense-in-depth style mechanisms on top of our basic design.
>
> In summary, our experiments demonstrate that a fixed security tensor yields consistent and significant safety gains across models and harmful categories. For stronger adaptive attacks, our preliminary analysis shows that simple adversarial overlays cannot reliably bypass our defense without severely degrading utility. Given the extremely small parameter size and easy updatability of the security tensors, we believe extending this mechanism to rapid patching and multi-tensor randomization for more complex adaptive settings is a highly promising direction for future work.  Thanks again for your comments.
>
>
> **[Weakness 6] How is the Harmless Rate (HR) in L298 evaluated—by matching refusal keywords? This needs clarification. If HR is based on refusal keyword matching, I would be curious to see results under semantic refusal detection.**
>
> ---
>
> Thank you for raising this question. We are happy to clarify that the Harmless Rate (HR) reported in L298 is evaluated purely via semantic refusal detection, not by matching refusal keywords. Each generated response on a malicious test sample is passed to a GPT-4o judge, which is prompted with a dedicated instruction designed to decide whether the reply should be classified as a “safety refusal.” Formally, the Harmless Rate (HR) reported in the paper is defined as the proportion of malicious visual test samples whose responses are judged by GPT-4o as semantic refusals under this procedure.
>
> Following your suggestion, we have added further details of this HR evaluation protocol, including the judging prompt, to Appendix A.1.10 in the revised version.  We greatly appreciate your pointing out this potential source of misunderstanding and hope this clarification addresses your concern.

---

> ### Author Response · Authors · 2025-11-20
> **Response to Reviewer  2kK7 - Part 4**
>
> **[Weakness 7] Do the results hold for fine-grained harmful categories (e.g., medical self-harm paraphernalia, non-gory violence, subtle hate symbols) or for text-in-image cases?**
>
> ---
>
> Thank you for raising this question. Our method is not designed as a classifier over a fixed set of visual categories, so its effect is not limited to “coarse, visually salient” harmful images. Conceptually, it should extend to fine-grained harmful categories (such as medical self-harm paraphernalia, non-gory violence, and subtle hate symbols) as well as text-in-image cases. Below we explain this from both a mechanistic and an empirical perspective.
>
> (1) Mechanistic perspective
>
> The proposed security tensors (visual $\delta_v$ and textual $\delta_t$) do not rely on any category labels. Instead, they work by shifting the joint image–text representations so that they activate the LLM’s existing safety subnetwork (safety layers). Therefore, as long as the image (or image–text pair) contains content that the LVLM can internally encode as harmful, the security tensors can help route the representation toward the pre-aligned safety subspace. This mechanism is not tied to a small set of “obvious” categories and is precisely why we observe consistent gains on multiple unseen harmful categories in our experiments.
>
> (2) Empirical perspective
>
> The harmful splits of VLGuard and MM-SafeBench used in our experiments already cover many fine-grained and implicit harmful pattern. Folders such as "hatefulMemes", "harm-p", and "bad_ads" in VLGuard contain rich instances of offensive language, incitement, scam prompts, and hate speech embedded as text within memes, advertisements, or screenshots. Plus, many images do not show explicit info but instead depict threatening poses, weapons in suggestive contexts, racial/religious targeting or potentially violent situations. On such “unseen, fine-grained, and implicit” harmful cases, our method still yields substantial improvements in Harmless Rate (HR).
>
> The per-category results reported in Table 1 of the main paper already aggregate over these more subtle patterns, so the reported improvements partially reflect the effect of our security tensors on such fine-grained and text-in-image cases. To make the effect on these hidden patterns more explicit, in the rebuttal phase we extracted a dedicated fine-grained and text-in-image subset from VLGuard, yielding 549 malicious samples in total.  On this subset, we measure how much the security tensors improve the base model’s Harmless Rate (HR). The absolute improvements (in percentage points) are in table 2:
>
> |            | Qwen-VL-Chat | LLaVA-1.5 | LLaMA-3.2-11B |
> | ---------- | ------------ | --------- | ------------- |
> | $\delta_v$ | 39.2%        | 36.1%     | 43.0%         |
> | $\delta_t$ | 38.6%        | 34.6%     | 42.4%         |
>
>  Table 2. Absolute improvement in Harmless Rate (HR, percentage points) on the fine-grained and text-in-image subset from VLGuard (hatefulMemes, harm-p, bad_ads) when applying visual-side (δ_v) or text-side (δ_t) security tensors, compared to the corresponding base models.
>
>  As shown, on this challenging subset with fine-grained, implicit, and text-in-image harmful content, our security tensors still provide large safety gains, improving HR by roughly 35–45 percentage points across three different LVLMs. This aligns with the broader unseen-category results and further confirms that the method is not restricted to a few coarse visual categories.
>
> ---
>
> Overall, we sincerely thank you for your comments for our articles and your dedication to reviewing our manuscripts!

---

> > ### Comment · Reviewer_2kK7 · 2025-11-27
> >
> > Thank you to the authors for the detailed response, which has resolved my concerns. I will keep positive. Good luck!

---

### Official Review · Reviewer_5hyW · 2025-10-30

**Soundness:** 3
**Presentation:** 3
**Contribution:** 3
**Rating:** 6
**Confidence:** 3

**Summary:**

This paper introduces Security Tensors, a novel cross-layer, prompt-based defense mechanism for large vision-language models (LVLMs). These tensors aim to transfer safety alignment mechanisms developed for text-based LLMs to the visual modality, enabling parameter-free, cross-modal safety enhancement. The approach optimizes both textual and visual tensors using a carefully curated dataset, aiming to improve security by enhancing harmful input rejection while maintaining the models’ original performance on benign inputs. Experiments on multiple LVLMs demonstrate strong safety improvements and generalization to unseen harmful visual categories with minimal performance degradation.

**Strengths:**

1. Quality: Sufficient experimental results are provided to demonstrate the method's effectiveness in improving safety while preserving the original models' performance. Additional evidence is also presented to explain the reasons behind this effectiveness.

2. Clarity: The paper is well-written and easy to follow.

3. Novelty: The proposed defense method is novel in the context of LVLMs. And to the best of my knowledge, the phenomenon of reusing the safety-guarding mechanism in the text-only decoder component is also new.

4. Significance: The proposed method generalizes to out-of-domain scenarios and can defend against more types of attacks, as shown in Table 1.

**Weaknesses:**

1. Quality: Currently, the performance degradation is measured only in the two proposed benchmarks, while in real-world applications, the models' other abilities are also important to track, such as mathematical reasoning [1] and commonsense reasoning [2].

2. Quality: The "safety layers" explanation is supported by evidence of cosine similarity values for different pair types. It could be strengthened if further activation patterns were available, such as the activation distribution in each layer.

### Reference

[1]: Lu, Pan, et al. "Mathvista: Evaluating mathematical reasoning of foundation models in visual contexts." arXiv preprint arXiv:2310.02255 (2023).

[2]: Yue, Xiang, et al. "Mmmu: A massive multi-discipline multimodal understanding and reasoning benchmark for expert agi." CVPR 2024.

**Questions:**

* Given Qwen-VL-Chat is a rather old model, what are the methods' performance in stronger VLM models such as Qwen-2.5-VL-instruct or Qwen-3-VL-instruct?

* Does the combination of both safe tensors yield further improvements?

* I am wondering if the proposed method can also generalize to categories such as deliberated elicited hallucinations [1].

### References

[1]: Han, Tianyang, et al. "The Instinctive Bias: Spurious Images lead to Illusion in MLLMs." EMNLP 2024.

**Details Of Ethics Concerns:**

If the proposed dataset is going to be released, ethics reviews are required for examining its content, especially the Safety Activation (SA) subset, which contains 400 manually crafted samples in categories of "Bloody", "Insult Gesture", "Guns", and "Porn".

---

> ### Author Response · Authors · 2025-11-20
> **Response to Reviewer 5hyW - Part 1**
>
> Many thanks for the insightful and helpful comments by the reviewer! We make the following point-by-point responses to your comments:
>
> **[Weakness 1]. Quality: Currently, the performance degradation is measured only in the two proposed benchmarks, while in real-world applications, the models' other abilities are also important to track, such as mathematical reasoning [1] and commonsense reasoning [2].**
>
> ---
>
> Thank you for your valuable comments! During the rebuttal phase, we additionally evaluated the impact of the security tensors $\delta_v$ and $\delta_t$ on mathematical reasoning and commonsense reasoning using the MathVista testmini dataset [1] and the MMMU benchmark [2]. For each model, we report performance in three configurations: the base model (without security tensors), with $\delta_v$, and with $\delta_t$. The results are summarized in Table 1.
>
> | | Qwen-VL-Chat | | | LLaVA-1.5 | | | LLaMA-3.2-11B | | |
> | - | - | - | - | - | - | - | - | - | - |
> | | Base | ST-$\delta_v$ | ST-$\delta_t$ | Base | ST-$\delta_v$ | ST-$\delta_t$ | Base | ST-$\delta_v$ | ST-$\delta_t$ |
> | MMMU | 35.9 | 36.0 | 36.2 | 36.4 | 35.4 | 35.8 | 50.7 | 50.2 | 50.6 |
> | MathVista (testmini) | 36.2 | 35.5 | 35.9 | 27.6 | 27.5 | 28.0 | 51.5 | 50.5 | 51.2 |
>
> *Table 1. Performance of different LVLMs on mathematical reasoning (MathVista testmini) and multi-discipline commonsense reasoning (MMMU) benchmarks, with and without the proposed security tensors $\delta_v$ and $\delta_t$.*
>
> From these results, we observe that introducing $\delta_v$ or $\delta_t$ at inference time does not degrade the models’ domain-specific reasoning abilities. Across all three LVLMs, the performance changes on both benchmarks stay within a small and reasonable range (approximately within ±1 absolute percentage point), with slight fluctuations in both positive and negative directions. This further supports the claim that our security tensors are essentially harmless to the models’ mathematical and commonsense reasoning capabilities while providing the desired safety alignment.
>
> We have included the above experimental results in Appendix A.1.8 of the revised version. Thank you once again for your thoughtful suggestions!
>
> [1]: Lu, Pan, et al. "Mathvista: Evaluating mathematical reasoning of foundation models in visual contexts." arXiv preprint arXiv:2310.02255 (2023).
>
> [2]: Yue, Xiang, et al. "Mmmu: A massive multi-discipline multimodal understanding and reasoning benchmark for expert agi." CVPR 2024.
>
>
>
> **[Weakness 2].  Quality: The "safety layers" explanation is supported by evidence of cosine similarity values for different pair types. It could be strengthened if further activation patterns were available, such as the activation distribution in each layer.**
>
> ---
>
> **[Answer to Weakness 2: Part 1/2]**
>
> Thank you for the constructive suggestion. In response, we have added a new set of quantitative neuron-level ablation experiments inspired by the concept of safety-critical neurons introduced in Wei et al. [1]. These experiments provide an additional perspective on how our security vectors activate and reuse the model’s existing safety alignment mechanisms in LVLMs, complementing the original “safety layers” analysis based on cosine similarity.
>
> Wei et al. [1] show, from a neuron-level perspective, that an LLM’s safety behavior is largely driven by a small, sparse, and identifiable subset of safety-critical neurons: masking only about 5% of these neurons can cause the refusal rate in text-only safety evaluations to drop from over 90% to around 10%. We reproduce and extend this analysis to LLaMA-3.2-11B-Vision to select roughly 5% of neurons via a set-difference procedure as the textual safety-critical neurons. On top of this, we design two ablation experiments on visually harmful inputs to test whether our security vectors meaningfully interact with this safety-critical neuron subset.
>
> **Experiment 1: Baseline LVLM without security vectors**
>
> On LLaMA-3.2-11B-Vision, we evaluate the same set of visually harmful inputs under two inference settings:
> (1) all neurons active;
> (2) the previously identified 5% safety-critical neurons are masked by zeroing their activations during the forward pass.
>
> On 200 test samples, the refusal rate decreases from about 24% (all neurons active) to about 13% (safety-critical neurons masked). This indicates that, in the original LVLM, these text-derived safety-critical neurons contribute roughly 10 percentage points to visual refusal, but they are far from fully activated: the model’s overall refusal rate for “image-embedded” harmful content remains low.

---

> ### Author Response · Authors · 2025-11-20
> **Response to Reviewer 5hyW - Part 2**
>
> **[Answer to Weakness 2: Part 2/2]**
>
> ---
>
> **Experiment 2: LVLM with our security vectors**
>
> We then repeat the same ablation procedure on the same visually harmful inputs, but now with our trained security vectors $\delta_t$ applied during inference. In this setting, we observe a dramatically different behavior:
>
> - With safety-critical neurons enabled, the refusal rate increases to about 84%.
> - Once we mask this 5% subset during inference, the refusal rate drops back down to about 16%.
>
> This large performance gap shows that, in the presence of our security vectors, the LVLM’s refusal behavior on harmful visual inputs becomes highly dependent on the same safety-critical neurons that are known to drive safety in text-only scenarios. In other words, the security vectors successfully activate and reuse these safety-critical neurons for the visual modality, leading to a substantial improvement in visual safety.
>
> In summary, the original layer-wise representation analysis in the paper already demonstrated, at the layer level, that our security vectors can reactivate safety layers that were originally sensitive only to harmful text. The new safety-critical neuron ablation experiments added in the rebuttal provide a complementary neuron-level perspective: they show that visual safety with security vectors explicitly depends on a sparse set of safety-critical neurons previously identified in text-only safety alignment. Together, these two lines of evidence offer a more rigorous and fine-grained validation that our security vectors effectively activate and reuse the LLM’s existing safety alignment capability, thereby strengthening the “safety layers” explanation beyond cosine similarity statistics alone.
>
> Thank you again for your valuable comments! We have incorporated the detailed procedures and results of this additional analysis into Appendix A.1.7 of the revised version.
>
> [1] Wei et al., Assessing the Brittleness of Safety Alignment via Pruning and Low-Rank Modifications, ICML 2024
>
> ---
>
> **[Question 1]. Given Qwen-VL-Chat is a rather old model, what are the methods' performance in stronger VLM models such as Qwen-2.5-VL-instruct or Qwen-3-VL-instruct?**
>
> ---
>
> Thank you for the helpful suggestion. During the rebuttal phase, we trained both the visual-side security tensor $\delta_v$ and the text-side security tensor $\delta_t$ on Qwen-2.5-VL-instruct and re-ran the full evaluation protocol used in the main paper. Concretely, $\delta_t$ is applied as a virtual-prefix tensor on the text tokens, while $\delta_v$ is directly added to the visual tokens. The evaluation results on Qwen-2.5-VL-instruct 7B are summarized in table 3:
>
> |               | security (HR) ↑ | benignness (FRR) ↓ | benignness (MM-Vet) ↑ |
> | ------------- | --------------- | ------------------ | --------------------- |
> | ST-$\delta_v$ | 82.2            | 1.5                | 66.0                  |
> | ST-$\delta_t$ | 80.4            | 1.0                | 66.7                  |
> | base model    | 17.7            | 0.5                | 67.1                  |
>
> Table 3. Safety (HR) and benignness (FRR, MM-Vet) of Qwen-2.5-VL-instruct 7B with and without security tensors.
>
> As shown, on Qwen-2.5-VL-instruct 7B our security tensors dramatically improve the Harmless Rate (from 17.7% to around 80%), while keeping the false rejection rate on benign inputs at a low level (≤ 1.5%) and maintaining MM-Vet scores within about 1 absolute point of the base model. This indicates that the method remains effective on stronger, more recent LVLMs.

---

> ### Author Response · Authors · 2025-11-20
> **Response to Reviewer 5hyW - Part 3**
>
> **[Question 2]. Does the combination of both safe tensors yield further improvements?**
>
> ---
>
> Thank you for the question. To directly address whether combining both safety tensors leads to further gains, we conducted additional experiments during the rebuttal phase.
>
> First, recall that $\delta_{v}$ and $\delta_{t}$ are each optimized independently in our framework: $\delta_{v}$ is trained in the visual pathway and $\delta_{t}$ is trained in the language pathway, each on our safety-training data. When evaluated separately, both tensors already provide strong safety activation effects. We then tested a simple combination strategy on LLaMA-3.2-11B-Vision by applying $\delta_{v}$ and $\delta_{t}$ together at inference time ($\delta_{v}$ + $\delta_{t}$). In this setting, we did not observe any improvement in safety performance compared to using each tensor alone; in some cases, the combined tensor even slightly degraded performance. A plausible explanation is that $\delta_{v}$ and $\delta_{t}$ are never co-optimized during training, so their effects can partially interfere with each other when they are naively added together at test time, leading to suboptimal safety activation.
>
> To further investigate this question, we additionally trained a single joint tensor $\delta_{v+t}$, which is injected into both modalities during training, so that its parameters are optimized under the combined visual–text safety objective. The resulting $\delta_{v+t}$ shows consistently better overall harmless rates than either $\delta_{v}$ or $\delta_{t}$ alone across different LVLMs. The comparison is summarized in Table 4:
>
> |                   | LLaMA-3.2-11B | Qwen-VL-Chat | LLaVA-1.5 |
> | ----------------- | ------------- | ------------ | --------- |
> | ST-$\delta_{v}$   | 84.2          | 64.5         | 49.5      |
> | ST-$\delta_{t}$   | 81.9          | 65.4         | 52.0      |
> | ST-$\delta_{v+t}$ | 87.3          | 69.1         | 56.9      |
>
> *Table 4. Overall harmless rate (%) on our safety benchmark when applying visual-only (ST-$\delta_{v}$), text-only (ST-$\delta_{t}$), and jointly trained multi-modal (ST-$\delta_{v+t}$) security tensors.*
>
> These results indicate that a jointly trained $\delta_{v+t}$ can indeed yield further improvements in safety performance. However, to be rigorous, we note that this improvement is not necessarily solely due to the “combination” of visual and textual effects per se. $\delta_{v+t}$ also has a larger effective parameter space than $\delta_{v}$ or $\delta_{t}$ individually, and part of the gain may come from this increased capacity rather than purely from the multimodal coupling mechanism. A more fine-grained disentanglement of these factors is an interesting direction that we plan to investigate in future work. Thank you again for raising this insightful question!

---

> ### Author Response · Authors · 2025-11-20
> **Response to Rviewer 5hyW - Part 4**
>
> **[Question 3]. I am wondering if the proposed method can also generalize to categories such as deliberated elicited hallucinations.**
>
> ------
>
> Thank you for the question! We would first like to clarify that the capability dimension targeted by our method is different from the Instinctive Bias phenomenon studied by Han et al. [1].
>
> Concretely, for an LVLM that has already been safety-aligned on the text side, its behavior can be roughly viewed as following two response pathways:
>
> (1) when harmful content is successfully detected, the model enters a “**safety pathway**” and produces a refusal response;
>
> (2) when the input is judged to be safe, the model stays on the “**normal pathway**” and generates a substantive answer.
>
> The practical problem we focus on arises when malicious information appears only in the image: in such cases, the LVLM often continues to follow the normal answering pathway and fails to trigger a safety refusal. By applying $\delta_v$ / $\delta_t$ in the input space, we aim to move “inputs containing harmful visual content” across the decision boundary from “normal pathway” to “safety pathway”, thereby reactivating the existing safety alignment ability that was originally sensitive only to harmful text.
>
> In contrast, the Instinctive Bias studied by Han et al. [1] is closer to a “factuality and robustness issue within the normal answering pathway.” Misleading but not explicitly harmful images cause the model to produce hallucinated or incorrect answers. The goal there is to move from “hallucination → incorrect answer” to “reduced hallucination → more accurate normal answer”, which does not involve switching between “safety” and “normal” pathways.
>
> Taken together, hallucination-related problems such as Instinctive Bias focus on improving factuality and robustness while staying within the same normal answering pathway, whereas our setting explicitly concerns when an input should exit that pathway and trigger a safety refusal instead. Accordingly, in this work, our data construction and loss design (SA/GB/TCB with CE/KL) are tailored to learning and sharpening the safety refusal boundary, rather than modeling or evaluating the same pathway's optimization.
>
> However, at the methodological level, it is indeed possible to extend the same “input perturbation / control vector” framework to hallucination-related behavior in future work, by introducing targeted annotations and loss functions specifically designed for hallucination control. This would correspond to learning a separate control vector aimed at reducing hallucinations along the normal answering pathway, which we see as an interesting and complementary extension beyond the scope of the present paper. We have added a brief discussion of this potential direction to Appendix A.1.8 in the revised version. Thank you again for raising this insightful point!
>
> [1] Han, Tianyang, et al. "The Instinctive Bias: Spurious Images lead to Illusion in MLLMs." EMNLP 2024.
>
> ---
>
>
>
> Finally, thank you sincerely for your valuable comments on our article and for your time and dedication in reviewing our manuscript. We hope that, building on your initial positive evaluation, the discussions we've made in response to your feedback will further strengthen your recommendation for acceptance. Thank you once again!

---

> ### Comment · Reviewer_5hyW · 2025-11-20
>
> Thanks for the detailed explanation. The response addresses all of my concerns, and I have raised my score.
>
> Regarding question 2, a fair comparison under the same capacity would be encouraged to be included in the future.

---

### Official Review · Reviewer_B9jr · 2025-10-31

**Soundness:** 3
**Presentation:** 3
**Contribution:** 3
**Rating:** 6
**Confidence:** 4

**Summary:**

The paper tackles a key gap in LVLM safety: text-only alignment does not reliably transfer to harmful visual inputs. The authors propose security tensors—trainable, input-level vectors injected either (a) in the text-embedding stream ($\delta_t$, virtual tokens between image and text embeddings) or (b) in the vision preprocessor feature space ($\delta_v$). These tensors are optimized offline (no model weights updated) using an asymmetric objective: cross-entropy to refusal on Safety Activation (SA) pairs (harmful image + benign text), and forward-KL distillation to baseline outputs on two benign sets—General Benign (GB) and a carefully designed Text-Contrast Benign (TCB) set whose prompts are syntactically similar to SA but paired with benign images. Evaluated on LLaMA-3.2-11B-Vision, Qwen-VL-Chat, and LLaVA-1.5 across VLGuard and MM-SafeBench hazards, the method substantially increases Harmless Rate (HR) with small increases in False Rejection Rate (FRR) and minor drops in MM-Vet Score. The paper also presents internal analyses suggesting that $\delta_t$ and $\delta_v$ reactivate “safety layers” in the LLM component (layers ~9–20), thereby bridging text-aligned safety to vision.

**Strengths:**

1: This paper introduces a method with low training cost that transfers the safety alignment capability of large language models (LLMs) to the image domain.

2: The paper conducts a layer-wise analysis to support its claim that the intrinsic safety alignment ability of the LLM is effectively activated by the proposed method.

**Weaknesses:**

1: The paper supports its claim that the safety alignment ability of the LLM is effectively activated through a layer-wise analysis. However, this evidence remains indirect. A quantitative ablation study would provide a more convincing and rigorous validation of this claim.

2: The paper lacks a clear explanation of how the learned security tensors activate the safety alignment ability of the LLM. Specifically, it remains unclear why changes in the latent representations trigger alignment behavior. Do the learned text-side security tensors carry any specific semantic meaning? Do the visual-side security tensors exhibit identifiable structural patterns? Providing such insights would significantly enhance the interpretability and credibility of the proposed mechanism.

**Questions:**

See Weaknesses.

---

> ### Author Response · Authors · 2025-11-20
> **Response to Reviewer B9jr - Part 1**
>
> Thank you sincerely for the insightful and constructive comments! We highly appreciate the valuable feedback received. In response, we offer a detailed point-by-point clarification to address each of the raised comments:
>
> **[Weakness 1]. The paper supports its claim that the safety alignment ability of the LLM is effectively activated through a layer-wise analysis. However, this evidence remains indirect. A quantitative ablation study would provide a more convincing and rigorous validation of this claim.**
>
> ---
>
> Thank you for the valuable suggestion! We analyze and demonstrate the effectiveness of the security vectors from both the layer-wise[2] and neuron-level[1] perspectives:
>
> **(1) Recap of layer-wise safety analysis (existing evidence in the main paper)**
>
> Following the methodology in Li et al. [2], we first identify a contiguous block of “textual safety layers” (approximately layers 9–20) in the language module of LLaMA-3.2-11B-Vision. These layers exhibit clear representational differences between harmful and benign text and are responsible for triggering refusal behavior. We then extend this analysis to the multimodal setting: when malicious information is present only in the image (with benign text), these safety layers—previously shown to be highly correlated with safety in the text-only setting—show almost no discriminative activation. This indicates that, in the original LVLM, the visual modality fails to effectively activate these safety layers.
>
> After introducing our security vectors, we observe that, under visually harmful inputs, these textual safety layers become discriminative again. This layer-wise analysis shows that our security vectors help the visual modality “plug into” the safety layers that were originally sensitive only to harmful text, thus providing indirect but meaningful evidence that the existing safety alignment ability is being reactivated.
>
> **(2) New neuron-level ablation based on safety-critical neurons**
>
> To more directly respond to your request for quantitative ablation, we introduce a second, independent line of analysis based on safety-critical neurons [1], and design neuron-level ablation experiments on top of it.
>
> Wei et al. [1] show, from a neuron-level perspective, that an LLM’s safety behavior is largely driven by a small, sparse, and identifiable subset of safety-critical neurons: masking only about 5% of such neurons causes the refusal rate in text-only safety evaluations to drop from over 90% to around 10%. We reproduce and extend this analysis to LLaMA-3.2-11B-Vision to select about 5% of neurons as textual safety-critical neurons. On top of this, we design two ablation experiments on visually harmful inputs to examine whether our security vectors make essential use of this safety-critical neuron subset.
>
> **Experiment 1: Baseline LVLM without security vectors**
>
> On LLaMA-3.2-11B-Vision, we evaluate the same set of visually harmful inputs under two inference settings:
>
> (1) all neurons active;
>
> (2) the previously identified 5% safety-critical neurons masked by zeroing their activations during the forward pass.
>
> On 200 test samples, the refusal rate decreases from about 24% (all neurons active) to about 13% (safety-critical neurons masked). This modest but noticeable drop suggests that, in the original LVLM, these text-derived safety-critical neurons are not fully activated by “image-embedded” harmful information: they contribute roughly 10 percentage points of refusal, but the overall visual refusal rate remains low.
>
> **Experiment 2: LVLM with our security vectors**
>
> We then repeat the same ablation procedure on the same visually harmful inputs, now with our trained security vectors $\delta_t$ applied during inference. In this setting, we observe a dramatically different pattern:
>
> - With safety-critical neurons enabled, the refusal rate increases to about 84%.
> - When we mask this 5% neuron subset during inference, the refusal rate sharply drops to about 16%.
>
> This large performance gap shows that, once our security vectors are applied, the model’s refusal behavior on harmful visual inputs becomes highly dependent on the same safety-critical neurons that drive safety in the text-only scenario. In other words, the security vectors successfully activate and reuse these safety-critical neurons for the visual modality, leading to a substantial improvement in visual safety.
>
> **(3) Summary**
>
> Together, these two complementary analyses offer a more convincing, quantitative, and ablative validation of our central claim that the existing safety alignment ability of the LLM is effectively activated and reused by our security vectors. We have included the detailed setup and additional results for this analysis in Appendix A.1.7 of the revised version.
>
> [1] Wei et al., Assessing the Brittleness of Safety Alignment via Pruning and Low-Rank Modifications, ICML 2024
>
> [2] Li et al., Safety Layers in Aligned Large Language Models: The Key to LLM Security,  ICLR 2025

---

> ### Author Response · Authors · 2025-11-20
> **Response to Reviewer B9jr - Part 2**
>
> **[Weakness 2]. The paper lacks a clear explanation of how the learned security tensors activate the safety alignment ability of the LLM. Specifically, it remains unclear why changes in the latent representations trigger alignment behavior. Do the learned text-side security tensors carry any specific semantic meaning? Do the visual-side security tensors exhibit identifiable structural patterns? Providing such insights would significantly enhance the interpretability and credibility of the proposed mechanism.**
>
> ---
>
> Thank you for the insightful question! Before answering in detail, we would like to clarify a background point: trainable continuous vectors (such as soft prompts, prefix / visual prompts, and universal perturbations) are generally not known to carry clear, human-interpretable semantics. On the text side, prior work has shown that soft prompts typically lie in a complex subspace of the high-dimensional embedding space; their internal structure is hard to reliably map back to specific words or phrases, and they are better understood as continuous control signals rather than “readable sentences” [1,2]. On the vision side, work on universal adversarial perturbations and visual prompts similarly suggests that these vectors often appear as weakly structured “noise” in pixel space, whose main effect is to shift features along certain directions in representation space, rather than corresponding to a recognizable object or shape [3,4].
>
> Under this understanding, we deliberately avoid over-interpreting our security tensors as having dictionary-like human semantics. Instead, we analyze them in a conservative way from both statistical and functional perspectives.
>
> On the visual side, we map $\delta_v$ back to the pixel space and compute its value histogram as well as visualizations. Specifically, the per-channel distributions are approximately zero-mean and roughly symmetric, resembling a near-Gaussian perturbation without clear local patterns or shapes. This is consistent with the behavior of “image-agnostic, small structured noise” rather than any semantically meaningful visual content. Consequently, we interpret $\delta_v$ as a universal safety perturbation: it does not encode specific objects or scenes, but acts as a global feature-space shift that, under harmful images, moves intermediate visual representations in directions that make downstream safety mechanisms easier to activate. The effect of $\delta_v$ in the pre-processed image space is illustrated in Figures 5 and 6 of Appendix A.1.3. In the rebuttal, we also provide the normalized value distributions and normal Q–Q plots of $\delta_v$ for both LLaMA-3.2-11B-Vision and LLaVA, at the anonymous link:
> https://anonymous.4open.science/r/ICLR2026_Rebuttal_Security_Tensor-BB87/distribution_analysis.jpg
>
> On the text side, we conduct a simple geometric analysis of the embeddings of $\delta_t$. First, the norm distribution of the virtual tokens is relatively balanced, and no position exhibits abnormally large magnitude. Second, when we project some of these virtual tokens onto the vocabulary embedding space and inspect nearest neighbors, we observe only weak correlations with a small number of refusal- or safety-related tokens (e.g., variants of “sorry”, “no”), while the overall neighbor set remains highly mixed. This suggests that $\delta_t$ should not be interpreted as a direct encoding of a readable refusal sentence. Instead, we view $\delta_t$ as a continuous safety-control prompt: its “semantics” are primarily functional, in the sense that it systematically amplifies the separability between harmful and benign representations in safety layers / safety-critical neurons when harmful visual input is present, while keeping its perturbation on benign inputs minimal.
>
> Overall, we will continue to investigate and refine the characterization of these tensors’ structure and behavior in future work. Thank you once again for your insightful comments！
>
> [1] Improving Complex Reasoning with Dynamic Prompt Corruption: A Soft Prompt Optimization Approach (Fan et al., ICLR 2025)
>
> [2] Towards Interpretable Soft Prompts (Patel et al., 2025)
>
> [3] How Deep Learning Sees the World: A Survey on Adversarial Attacks & Defenses (Costa et al., IEEE Access 2024)
>
> [4] DA-VPT: Semantic-Guided Visual Prompt Tuning for Vision Transformers (Ren et al., CVPR 2025)
>
> ---
>
> **We sincerely appreciate your valuable feedback and the time you dedicated to reviewing our manuscript. Building on your initial positive assessment of this work, we hope that the discussions and improvements we have made in response to your comments will further strengthen your recommendation for acceptance. Thank you once again for your thoughtful and constructive input!**

---

### Author Response · Authors · 2025-12-02
**Summary of Reviewer Feedback and Rebuttal Resolution**

Dear AC, SAC, and PCs,

Thank you for taking the time to evaluate our submission under the current circumstances. Below we briefly summarize how the reviewers assessed the paper and how their main concerns were addressed.

**Positive feedback from reviewers.** Our paper proposes *security tensors*, a parameter-free, plug-in defense that reuses and activates the text module’s existing safety mechanisms for the visual modality in LVLMs. Reviewers broadly agreed that this idea is both novel and practically relevant. In particular:

* 5hyW emphasized that the defense is novel in the LVLM setting and that the idea of reusing the text-only decoder’s safety mechanism is new.
* B9jr, 2kK7, and ZYrM highlighted that our method works with low training cost and causes almost no performance drop on benign tasks.
* B9jr, 5hyW, and 2kK7 also appreciated the detailed experiments and layer-wise analysis showing that security tensors can activate the LVLM’s text-side “safety layers.”

**Rebuttal additions and resolution of concerns.** During rebuttal, two reviewers (5hyW and 2kK7) engaged in discussion and explicitly stated that our rebuttal successfully addressed their concerns; reviewer 5hyW, in particular, raised their score from 6 to 8 before the rollback.

Although the remaining reviewers did not participate in the discussion phase before the rollback, many of their concerns overlapped with those of 5hyW and 2kK7, and were already addressed in the responses that these two reviewers explicitly endorsed. In addition, we carefully added several new analyses and experiments aimed at the residual concerns, including but not limited to: (i) feature-level analysis of the visual and textual security tensors, (ii) experiments on LVLMs of different sizes demonstrating the generality of security tensors, (iii) evaluations combining security tensors with other pre-processing defenses, showing a more-than-additive (“1+1>2”) effect, and (iv) a detailed analysis of computational cost further confirming that security tensors provide a lightweight defense. Taken together, we believe these efforts have also effectively addressed the concerns raised by the remaining reviewers who did not join the discussion.

We understand that the scores have now been reverted, but we hope this context is helpful in assessing how the reviewers’ views evolved over time. We are grateful for your effort in navigating a difficult situation and for considering both the written reviews and the discussion dynamics. Finally, we confirm that all interactions with reviewers occurred exclusively via OpenReview; we have never contacted any reviewer or AC outside the system.

Thank you again for your time and consideration.

— The authors

---

### Meta-Review · Area_Chair_usJn · 2025-12-07

**Summary:**

This paper introduces a novel defense paradigm against jailbreak attacks on LVLMs by optimizing an input-level security tensor to reliably trigger refusal on harmful prompts. Reviewers found the core idea to be original, lightweight, and practically effective, highlighting the strong safety improvements, preservation of benign capabilities, and thorough analyses. Nonetheless, several concerns remain, including limited interpretability of the mechanism, incomplete evaluation breadth (e.g., reasoning tasks, model scaling, adaptive adversaries), reliance on the base model’s existing safety alignment, insufficient computational analysis, and questions regarding dataset size, refusal-template diversity, and evaluation design. Although the rebuttal provided substantial new experiments and clarifications, not all issues were fully resolved.

**Reviewer Concerns:**

**Addressed**: Reviewers 5hyW and 2kK7 explicitly confirmed that the rebuttal resolved their concerns, including interpretability, benign-task evaluation, dataset construction, template diversity, and unseen-category generalization.

**Outstanding**: Some concerns from reviewer ZYrM remain only partially addressed, especially model-scaling analysis depth, robustness under adaptive attacks, dependency on base alignment, and more systematic computational and defense-in-depth evaluation.


After reviewing the paper, the reviewer comments, and the author rebuttal, the AC decided to reject the paper, despite the initially high rating of 6664. The reasons are as follows:

1. The method is not clearly differentiated from existing techniques such as prompt tuning, test-time prompt tuning, or guard-model–based safety filtering. Conceptually, it behaves very much like a prompt-tuned guard mechanism that detects and rejects harmful prompts, yet the authors do not acknowledge this resemblance and instead frame the approach as a novel conceptual contribution.

2. The proposed method only produces rejections. Modern safety research increasingly emphasizes the dual goals of helpfulness and harmlessness, rather than simply refusing. A mechanism that exclusively rejects harmful prompts essentially replicates the behavior of a guard model, yet the paper presents it as a cross-modal alignment bridging. This conceptual mismatch weakens the framing and raises questions about the actual contribution. I believe this work requires a more in-depth and well-grounded functionality and mechanistic understanding before it is ready for publication.

3. The experimental evaluation has major limitations. No commercial models were evaluated, and the three datasets used are relatively small with limited diversity and coverage (in both topics and attack types). This makes it difficult to assess the generality of the proposed methods against the broad spectrum of recent attacks. The paper does not compare against broader guard models or safety defenses that are standard in this line of work.

**Reviewer Scores:**

2kK7: 8
B9jr: 6
5hyW: 8
vS6s: 4

---

### Decision · Program_Chairs · 2026-01-26

Reject